

# Acoustic and optical methods to infer water transparency at the Time Series Station Spiekeroog, Wadden Sea (southern North Sea)

Anne-Christin Schulz[1], Thomas H. Badewien[1], Shungudzemwoyo P. Garaba[2], and Oliver Zielinski[1]

[1]Institute for Chemistry and Biology of the Marine Environment, Carl von Ossietzky University of Oldenburg, Schleusenstr. 1, 26382 Wilhelmshaven, Germany.
[2]Department of Marine Sciences, Avery Point Campus, University of Conneticut, 1080 Shennecosset Road, Groton, CT 06340, USA.

*Correspondence to:* Anne-Christin Schulz (anne.c.schulz@uni-oldenburg.de)

**Abstract.** Water transparency is a key indicator of optical water quality that is driven by suspended particulate and dissolved material. In this study we carried out an intercomparison of observations related to water transparency, determine correlations among the measured parameters and demonstrate the utility of both acoustic and optical tools in monitoring water transparency. The data set used here is from the operational Time Series Station Spiekeroog located at a tidal inlet of the Wadden Sea. An

Acoustic Doppler Current Profiler was used to obtain acoustic measurements in the water column. Optical observations were determined using a set of three radiometers above water to collect radiometric quantities and a turbidity sensor within the water column. Bio-fouling was identified as a source of anomaly in turbidity measurements. We observed significant correlations between in-situ optically measured turbidity and derived turbidity from above water color sensing and acoustic backscattering strength. These findings underline that both optical and acoustic measurements can be reasonable proxies of water transparency

with the potential to mitigate gaps and increase data quality in long-time observation of marine environments.

Keywords: Acoustic backscatter, turbidity, Forel-Ule-Index, Time Series Station, remote sensing reflectance.

## 1 Introduction

Over the last years, the importance of water quality and environmental awareness has risen for the public, scientists and policy

makers (WFD, 2000; OECD, 1993; Borja et al., 2013). In order for us to better understand water quality, it is important to gather environmental information from different tools on platforms at varying temporal and spatial coverage (Zielinski et al., 2009; Pearlman et al., 2014). The information from different tools ought to be comparable for a better understanding of environmental dynamics that drive water quality. However, water quality is an abstract term that is used widely to describe the state of water according to its ecological, chemical, optical or morphological properties; each of these having a set of related

variables that are used as indices. One of the key indicators for water quality is the water transparency (CRWN, 2012). Two main aspects of water clarity (Davies-Colley and Smith, 2001) or water transparency (Wilson, 2010) are the light penetration and the visual clarity (Kirk, 1988). Both are strongly affected by dissolved and suspended material within the water (Kirk, 1985) and the optical properties of the water.



Optical water transparency is determined from optical observations that involve using the human eye as a tool or methods that replicate the human eye sensing approach (Moore et al., 2009). In general, water transparency has been determined for decades as the tools needed are easy to use, fast, inexpensive and robust. Typical optical observations provide information about the light availability in the water column which can be translated into water transparency. Turbidity is one such measurements

which refers to a relative index of water cloudiness influenced by the inherent dissolved and particulate material (Kirk, 1985; Moore, 1980). Remote sensing reflectance ($R_{RS}$), an essential climate variable, is a proxy for the intrinsic color of water and optically active constituents of water (Garaba et al., 2015; Garaba and Zielinski, 2013; GCOS, 2011; Watson and Zielinski, 2013). The natural color of water, driven by the optically active constituents of water and environmental conditions, can be distinguished using a standard Forel-Ule color scale. The Forel-Ule color scale numerically categorizes natural waters from 1

(indigo-blue) to 21 (cola brown), and this information can also be derived from $R_{RS}$ information (Garaba et al., 2014; Wernand and van der Woerd, 2010; Garaba et al., 2015).

In more recent years, acoustic backscatter has been used to estimate abundance and distributional patterns of suspended matter (Thorne and Hanes, 2002; Deines, 1999; Thorne et al., 1991). Acoustics is one of a number of technologies advancing our capabilities to probe sediment processes (Thorne and Hanes, 2002; Voulgaris and Meyers, 2004). Acoustic backscatter signals

provide information about the suspended material in a given water body and enable to record the changes over a long time scale. An Acoustic Doppler Current Profiler (ADCP) measures non-intrusive and three-dimensional, making it a very powerful tool for examining small-scale sediment transport processes (Thorne and Hanes, 2002; Schulz et al., 2015).

In coastal and estuarine regions, the composition and concentration of suspended material is highly variable (Winter et al., 2007; Fugate and Friedrichs, 2002). Fragile flocculants change their characteristics over short and over long time scales due to

hydrodynamic forcing conditions among them currents, turbulence and tides (Vousdoukas et al., 2011; Fugate and Friedrichs, 2002; Burchard and Badewien, 2015). Depending on whether optical methods (White, 1998; Sutherland et al., 2000) or acoustic methods (Voulgaris and Meyers, 2004; Fugate and Friedrichs, 2002) are used, the scattering properties of sediment differ. Optical measurements are more sensitive to fine sediment, while acoustic measurements are more sensitive to coarser material, depending on the operating frequency (Gartner, 2004). Thus Winter et al. (2007) concluded that the combination of differ-

ent instruments reveal different aspects of SPM dynamics. Comparisons of acoustic and optical sensors for measurements of suspended sediment concentrations performed under laboratory conditions showed that most in-water sensors have a linear response under bimodal and randomly sorted suspended sediments (Vousdoukas et al., 2011).

In this work, we therefore aim to find out if the information from optical water quality variables is related to the backscatter signal. The assumption is based on the fact that all these observations provide information about the inherent suspended partic-

ulate and dissolved material and therefore can be practical indicators of water transparency. We also evaluate to which extent combining acoustic and optical techniques are suited for environmental monitoring. We discuss additionally, whether qualitative and quantitative information gathered at operational long time series observatory platforms helps identifying indicators of change within natural waters. Comparing different tools for analysing water quality may also prove valuable for assessing precision as well as accuracy of measurements. At the same time, this approach also helps, understanding associations among the



observed variables. This in turn may enable to close gaps in relevant information concerning dynamics of suspended material in the water column when individual instruments fail.

## 2 Materials and methods

### 2.1 Study area

The Time Series Station Spiekeroog (TSS, Fig. 1) is an operational multidisciplinary autonomous observatory located (53° 45.016' N, 007° 40.266' E) in a tidal channel between the islands of Langeoog and Spiekeroog (Badewien et al., 2009). These islands are part of an island barrier system in the East Frisian Wadden Sea, southern North Sea, which belongs to the UNESCO World Natural Heritage sites. The region is part of an extended North Sea tidal flat system with shallow water depths ranging from 0 to 20 $m$. The typical water depth is 13.5 $m$ with tidal range of about 2.7 $m$ at the site of the Time Series Station

Spiekeroog (Holinde et al., 2015). Usual current velocities during ebbing and flooding periods reach values of up to 2 $m/s$. The distribution of suspended particulate and organic material is strongly influenced by the semi-diurnal tidal currents as well as waves driven by wind (Bartholomä et al., 2009; Reuter et al., 2009; Badewien et al., 2009). These strong and rapid dynamics around TSS make it a valuable study area for biogeochemical and physical investigations in transitional waters from land to the open sea.

### 2.2 Sampling and analysis

Three radiometers (TriOS, Germany) continuously measure with a 5 minute interval, hyperspectral radiance (2 x RAMSES ARC) and irradiance (1 x RAMSES ACC) to assess remote sensing reflectance (R$_{RS}$). The radiometers are mounted to the TSS at 24 $m$ above the seafloor (approx. 12 $m$ above NN). The radiometric quantities are corrected for environmental perturbations (Busch et al., 2013; Garaba et al., 2015, 2012) and are transformed into Forel-Ule color indices that can be matched to the

intrinsic color of water (Wernand, 2011). Detailed information on the processing of these measurements is presented in an earlier study (Garaba et al., 2014). A submerged ECO FLNTU (WETlabs, USA) sensor continuously measures turbidity at 12 $m$ above the seafloor with 1 minute interval. The ECO FLNTU sensor samples turbidity data with optical backscattering at a wavelength of 700 $nm$. The sensitivity of the sensor is 0.01 $NTU$ and the typical range of turbidity is between 0 $NTU$ and 25 $NTU$. Because of problems associated with bio-fouling, turbidity measurements at the TSS might not be at all times reliable.

In order to compare optical data to other variables, data were taken from a period directly after cleaning the ECO FLNTU sensor (low water, 28 August 2013) from bio-fouling. Figure 2 shows the turbidity data in $NTU$ over a time period from 28 August 2013 until 02 September 2013. The blue points show the in-situ raw data, limited between 0 $NTU$ and 25 $NTU$. The red points represent quality checked turbidity data. These turbidity data were measured near the surface of the water column at a fixed height above the seabed, therefore these data can directly used for the comparison.


A bottom-mounted (1.5 $m$ above the seafloor), upward looking 1200 $kHz$ ADCP (Teledyne RD Instruments Workhorse Sen-





tinel, USA) estimates the current velocity using the Doppler effect in three dimensions. The ADCP is installed at a distance of about $12\ m$ towards North-north-west from the pole of the station. We receive data over the entire water depth with a vertical resolution of $0.20\ m$ (bin size) and a temporal resolution of 5 minutes (measurements are averaged over 45 pings in 5 minute bursts). Because the ADCP has a beam angle of $20°$ and a tilted orientation with pitch $\sim 19.39°$ and roll $\sim 17.96°$, the maximal

range $R_{max}\ [m]$ of acceptable data is given by

$$R_{max} = D\cos(\phi) \tag{1}$$

where $D$ is the distance from ADCP to the surface in $[m]$, and $\phi$ is the angle of the beam relative to vertical $[°]$. The resulting blank space near the surface reached values between $3.0\ m$ to $3.5\ m$ (see Fig. 3). In order to fill this gap, we extrapolated the acceptable backscatter data from $R_{max}$-depth to the sea surface layer.

In different tidal phases (flood, ebb, slack water) the backscatter signal shows different profiles over depth. Figure 4 displays that the profiles at slack water (high / low water - light blue, orange, violet) have similar shapes. Also, the profiles at maximum velocity (flood / ebb - green, blue, dark yellow, brown) show similar shapes.

To extrapolate the backscatter data to the surface layer, various curve fitting techniques using MATLAB R2015a, Curve Fitting Toolbox (MathWorks, USA) were implemented. These methods were (i) the exponential fitting method ('exp', equation:

$a \cdot exp(b \cdot x)$), (ii) the polynomial fitting method ('poly', equation: $p1 \cdot x + p2$), (iii) the power fitting method ('power', equation: $a \cdot x^b$) and (iv) constant extrapolation (using the last reliable data value as the surface value). Comparisons between these methods are shown in Fig. 5. The top left graphic shows a profile during low water (violet in Fig. 4) and during flood (blue in Fig. 4) with different extrapolations to the surface layer. The top right graphic shows the extrapolation results. Here, the data derived in the near bottom range up to $4\ m$ are excluded since it is highly influenced by strong currents during ebb and flood.

We assume that the distribution of suspended matter near the surface layer is homogeneous because of turbulence and wind influence.

The best results for extrapolation over different existing data sets (extrapolation with complete (wwc) and reduced water column (rwc)) are given by the exponential extrapolation ('exp'). The $R^2$ values over the entire period are good with $R^2 > 0.99$ (see Fig. 5, bottom). Figure 6 shows the resulting signals at the surface layer. To compare backscatter data with the other

parameters, the exponential fitting method and the extrapolations with constant values ($BS_{Ex,exp}$ and $BS_{Ex,const}$) were used. The latter was used to fulfil the assumption of homogeneity in the top layer of the water column.

The acoustic backscatter signal is used to quantitatively determine suspended matter and therefore relates to turbidity (Deines, 1999; Lohrmann, 2001; Schulz et al., 2015).

One aim of this study is the inter-comparison of measurements from different tools to understand correlations among the observed variables and to mitigate gaps in relevant information concerning changes in suspended material or water transparency in the water column, e.g. when instruments fail (Oehmcke et al., 2015). Vousdoukas et al. (2011) found a linear response between SPM data obtained by optical and acoustic sensors in laboratory conditions under various suspended sediment conditions. Our early investigations estimated a linear relation between turbidity and Forel-Ule-Index (Garaba et al., 2014) and





a polynomial function of second order between turbidity and acoustic backscatter signal (Schulz et al., 2015). For this study, we consider for the first time all data sets of the different measurement approaches (acoustic and optical, in-water and above water) to evaluate their utility for environmental in-situ monitoring of information on water transparency variables. In order to confirm correlations, we apply the Spearman rank correlation test.

## 3   Results and discussion

A time series of the acoustic and optical measurements from the 29.08.2013 is presented in Fig. 7. Visual inspection of the data suggested that there was a moderate correlation. We assume it is because of the position of the sensors, above the sea surface, submerged near the sea surface and submerged near the seafloor (an overview of the fields of view is shown in Fig. 8). Additionally, the lack of a strong correlation may be due to the different scatter behaviour of suspended sediment (White, 1998; Sutherland et al., 2000; Voulgaris and Meyers, 2004; Fugate and Friedrichs, 2002) depending on whether optical or acoustic methods are applied. Optical measurements are more sensitive to fine sediment and are wavelength dependent, while acoustic measurements are more sensitive to coarser material, depending on the operating frequency (Gartner, 2004). Both depend on the amount of suspended sediment in the water column. Badewien et al. (2009) measured a range of particle sizes from 1.25 to 26.9 $\mu m$ (radius) at the same site. Therefore, mainly fine sediment concentrations are expected. A modelling study by Stanev et al. (2007) showed different sediment concentrations and dynamics for fine SPM (mud, $d_{mud}$ = 63 $\mu m$) and sand ($d_{sand}$ = 200$\mu m$) for this Wadden Sea area. Depending on the specific location, the dynamics of the different sediment types (fine or coarse) acted differently dependent on the tidal signal. The dynamics of all data sets correspond well to the observed tidal signal. Furthermore, we presume these instruments provide a reasonable proxy for the suspended material which is comparable. In order to confirm the correlations we applied the Spearman rank correlation test (for two time periods of different length). Results of the Spearman rank correlation are shown in table 1. To compared the data at nearly the same sampling target, we extrapolated the backscatter signal towards the sea surface area. Two of these extrapolated variables ($BS_{Ex,const}$ and $BS_{Ex,exp}$) are used for the Spearman rank correlation test. The correlation coefficient between the data sets increases from moderate ($\rho_{Spearman} > 0.4$ and $\rho_{Spearman} < 0.6$) to strong ($\rho_{Spearman} > 0.6$ and $\rho_{Spearman} < 0.8$). The correlation between Forel-Ule-Index $FUI$ and turbidity $TRB$ is even very good ($\rho_{Spearman} > 0.8$). For further investigations, we use the constant extrapolated backscatter signal $BS_{Ex,const}$.

The Forel-Ule color scale is an inexpensive and longstanding tool used to determine intrinsic color of water (Garaba et al., 2015; Garaba and Zielinski, 2015). In a previous study it was shown that the Forel-Ule-Index (FUI) can be used to accurately derive turbidity (Garaba et al., 2014). We therefore evaluate its potential in providing information about suspended material, which in turn can be compared to information derived from backscatter signals (Fig. 9). We made comparisons with data obtained from different tidal phases (ebb, flood and slack water periods) and observed correlations between $\rho_{Spearman} = 0.34$ (for high tide) and $\rho_{Spearman} = 0.81$ (low tide) (all results are shown in table 2). Figure 10 shows a comparison between the backscatter signal and the turbidity data, which are separated into the tidal phases: ebb (green), flood (red), high waters (dark blue), low waters (light blue). The Fig. shows also the comparison between the backscatter signal from the ADCP which





was constantly extrapolated to the sea surface $BS_{Ex,const}$ versus Forel-Ule-Index $FUI$ on the left and $BS_{Ex,const}$ versus turbidity data $TRB$ on the right. Spearman rank correlation tests were also applied to signals at different tide phases (ebb, flood and slack water periods) and results are presented in table 2. The correlations between backscatter data $BS_{Ex,const}$ and Forel-Ule-Index $FUI$ are weak at high tide ($\rho_{Spearman} = 0.34$) and mostly moderate ($\rho_{Spearman} > 0.4$ and $\rho_{Spearman} < 0.6$).

The correlations between backscatter data $BS_{Ex,const}$ and turbidity $TRB$ are weak at high tide and flood and otherwise strong ($\rho_{Spearman} > 0.6$ and $\rho_{Spearman} < 0.8$).

## 4   Conclusions

The results of this study show that bio-fouling decreases the data quality of in-water optical measurements of turbidity within short time periods. Hence, it is important to find an approach to improve the monitoring over time and increase the robustness

of the turbidity results. This study demonstrates that bottom-mounted ADCP measurements, which are hardly influenced by bio-fouling, is a suitable alternative to overcome the problem. We found that using the backscatter signal and Forel-Ule-Index both yield reliable results, broadening the work of Garaba et al. (2014). Our results regarding the correlation between backscatter signal and turbidity agree well with investigations of Schulz et al. (2015). The linear responses between the different sensors types found in this study agrees with previous results from laboratory investigations (Vousdoukas et al., 2011).

We have also shown that data sets from different measurement principles (optical and acoustic) are comparable and complementary. This is despite of the position of the sensors, above the sea surface, submerged near the sea surface and submerged near the seafloor and the well-known fact that the scattering properties of particles derived from both methods differ (White, 1998; Sutherland et al., 2000; Voulgaris and Meyers, 2004; Fugate and Friedrichs, 2002).

Thus, these methods can be utilized as an affordable tool and from different platforms to monitor environmental processes. On

a qualitative level, using the Forel-Ule-Index, derived from radiometer measurements, is a powerful tool as much as data sets derived from ADCP measurements. Our investigations also underline that long term observatories are key in understanding the marine environment because short-term studies only allow for a limited view on the considered dynamics.

Summarizing we found out that the information from optical water quality variables is related to the acoustic backscatter signal. We also evaluated the utility of acoustic and optical technology in environmental monitoring to gather qualitative and

quantitative indicators of change within natural waters taking advantage of operational long time series observatory platforms. The goals of this study was to perform an inter-comparison of measurements from different tools, to understand correlations among the observed variables, and to develop methods geared at mitigating gaps in relevant information about changes in water transparency in the water column when instruments fail.

*Acknowledgements.* We would like to thank Axel Braun, Helmo Nicolai, Gerrit Behrens and Waldemar Siewert for their ongoing technical

assistance and support in all our experimental work and the maintenance of the Time Series Station Spiekeroog. We also thank Constanze Böttcher for English language editing. Thanks to Nick Rüssmeier for CAD illustration. The funding for the technical renewal of the time series station by the Coastal Observation System for Northern and Arctic Seas (COSYNA) project is gratefully acknowledged.



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





**Table 1.** Spearman rank correlation results of the backscatter data from the ADCP $BS_{EX,const}$ and $BS_{EX,exp}$ for one day and for a longer time period (29.08.2013 - 02.09.2013) as well as the estimated Forel-Ule-Index $FUI$ and the turbidity $TRB$.

| variables | $\rho_{Spearman}$ one day | $\rho_{Spearman}$ longer period | p-value one day | p-value longer period |
|---|---|---|---|---|
| $BS_{EX,const}$ vs. $TRB$ | 0.78 | 0.50 | <0.001 | <0.001 |
| $BS_{EX,exp}$ vs. $TRB$ | 0.67 | 0.42 | <0.001 | <0.001 |
| $BS_{EX,const}$ vs. $FUI$ | 0.58 | 0.52 | <0.001 | <0.001 |
| $BS_{EX,exp}$ vs. $FUI$ | 0.48 | 0.44 | <0.001 | <0.001 |
| $FUI$ vs. $TRB$ | 0.88 | 0.85 | <0.001 | <0.001 |

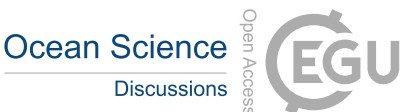

**Table 2.** Spearman rank correlation results of the backscatter data from the ADCP $BS_{EX,const}$, the estimated Forel-Ule-Index $FUI$ and the turbidity $TRB$ - separation tidal phases.

| variables | tide phase | $\rho_{Spearman}$ | p-value |
|---|---|---|---|
| $BS_{Ex,const}$ vs. $FUI$ | ebb | 0.45 | <0.001 |
| | flood | 0.52 | <0.001 |
| | high tide | 0.34 | 0.06 |
| | low tide | 0.81 | <0.001 |
| $BS_{Ex,const}$ vs. $TRB$ | ebb | 0.71 | <0.001 |
| | flood | -0.34 | <0.001 |
| | high tide | 0.40 | 0.0014 |
| | low tide | 0.77 | <0.001 |




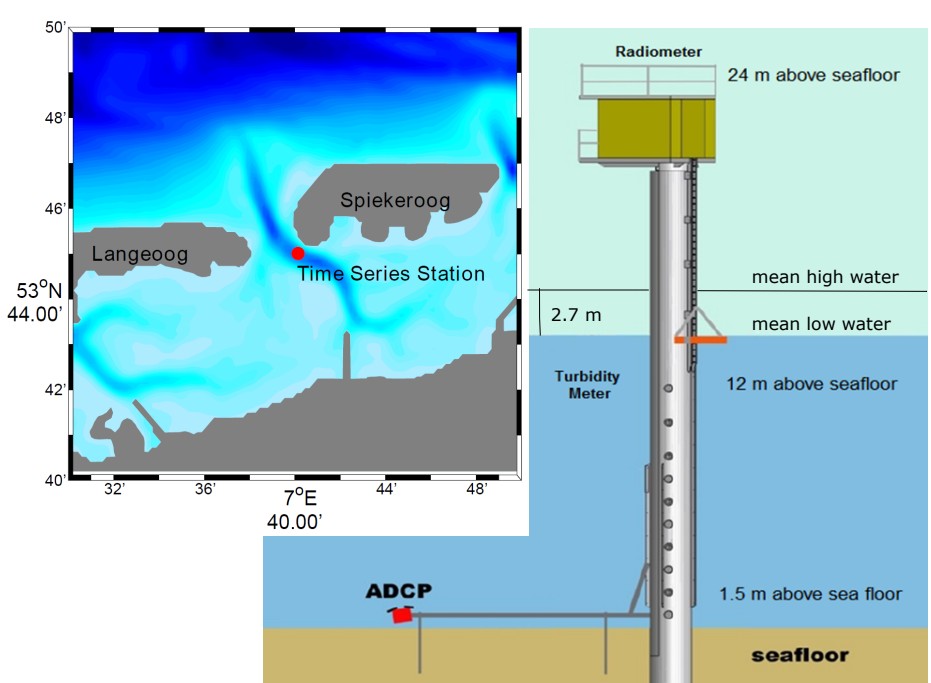

**Figure 1.** Schematic of the Time Series Station Spiekeroog showing the position of the radiometers (24 $m$), turbidity meter (12 $m$) and ADCP (1.5 $m$ above the seafloor). The typical water depth is 13.5 $m$ with tidal range of about 2.7 $m$ between mean high and low water. The insert shows the location of the Time Series Station where the colors indicate the water depth at high water.



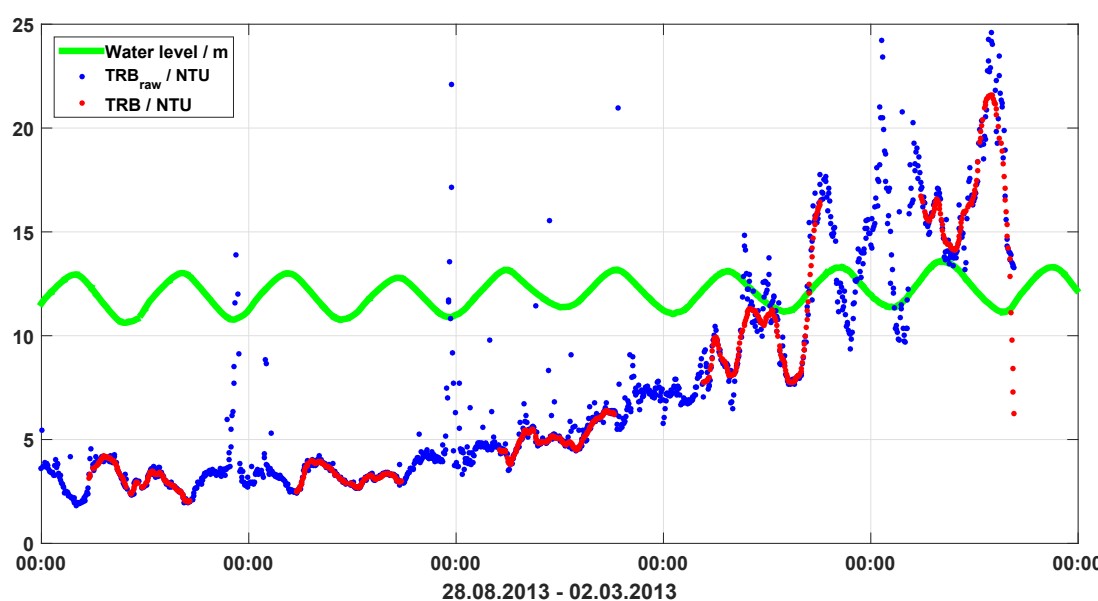

**Figure 2.** Turbidity data in $NTU$ from 28 August 2013 until 02 September 2013, limited in range (0 - 25 $NTU$), blue: raw data, red: quality checked data and green: water level in $m$.



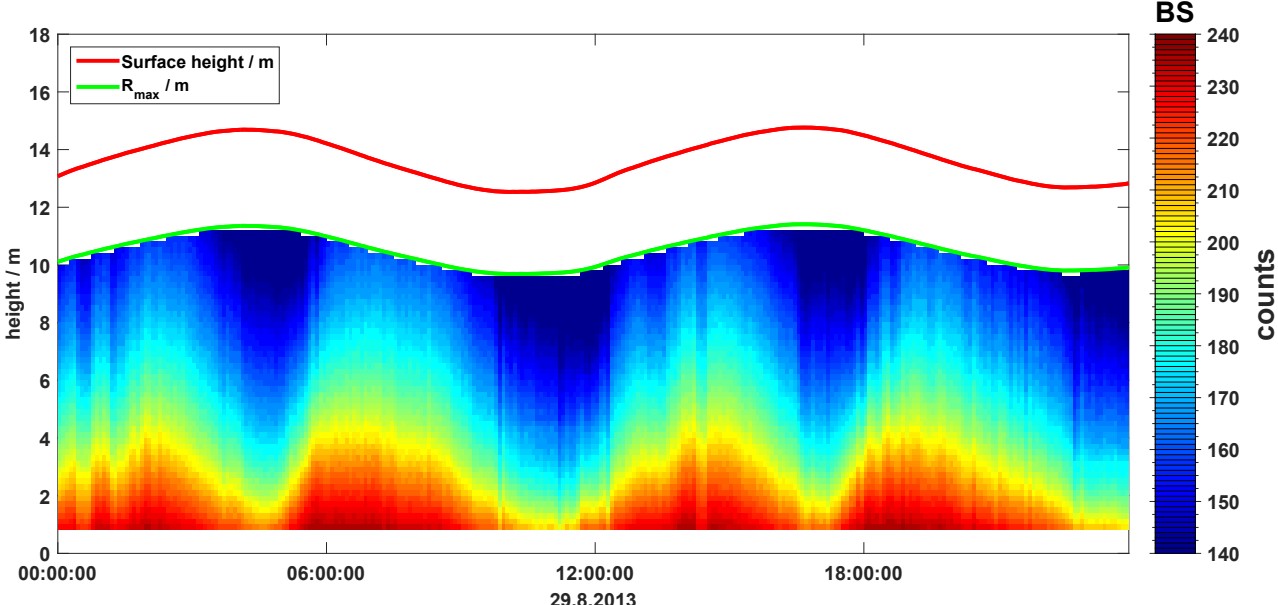

**Figure 3.** An example of backscatter signal in counts, acceptable backscatter data until $R_{max}$. Green line: $R_{max}$-depth / $m$; red line: sea level (height / $m$) observed on 29 August 2013.



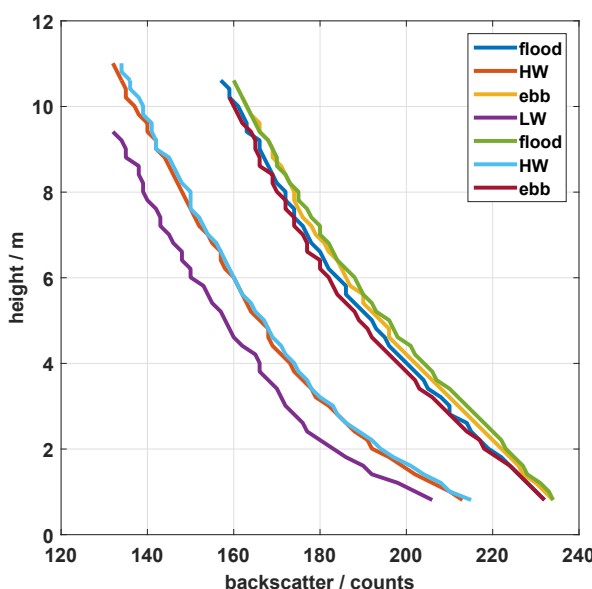

**Figure 4.** Example backscatter profiles over depth in counts observed on 29 August 2013 at different tidal phases: flood, ebb, slack water (high water (HW), low water (LW)).



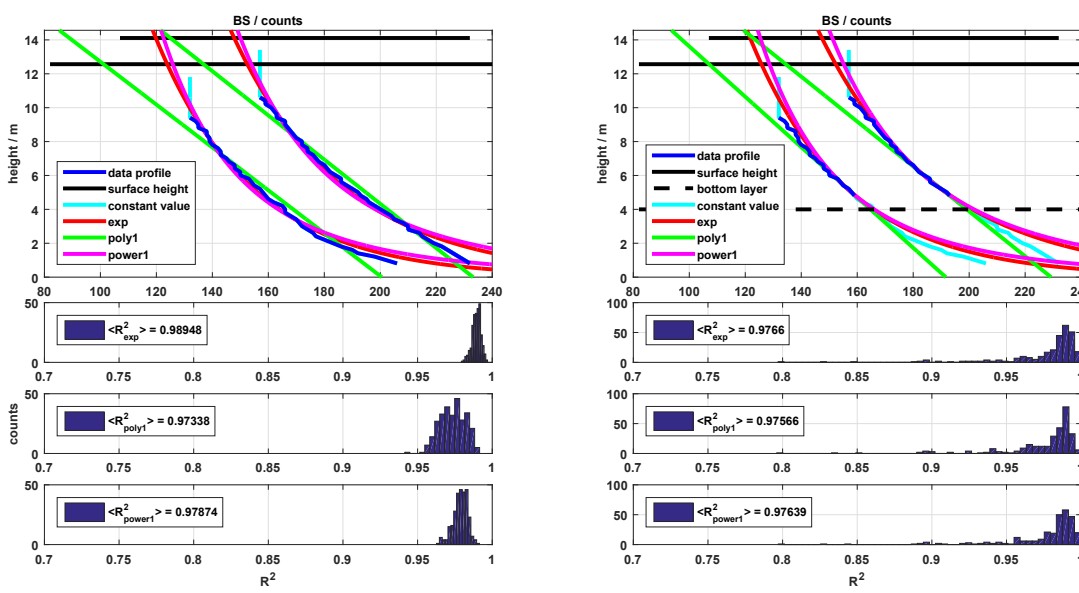

**Figure 5.** Selected backscatter profiles during low water and during flood; top left: extrapolation through the whole water column; top right: extrapolation through the reduced water column. The colored profiles show the results of the different extrapolation methods (cyan: constant extrapolation; red: exponential extrapolation; green: polynomial extrapolation; magenta: power extrapolation). Black line: surface layer and black dotted line: Lower layer. Bottom graphics: the corresponding $R^2$ values for the entire period; left: for the whole water column, right: for the reduced water column.



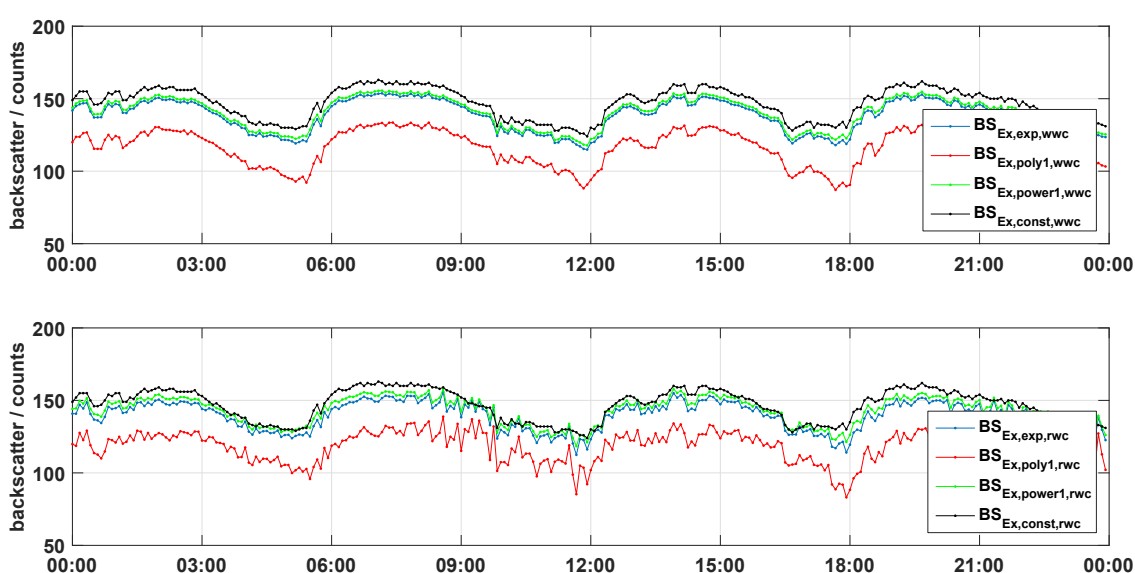

**Figure 6.** Comparison of extrapolated data sets towards the surface layer as a time series: whole water column (wwc) and reduced water colum (rwc) (black: constant extrapolation; blue: exponential extrapolation; red: polynomial extrapolation; green: power extrapolation).





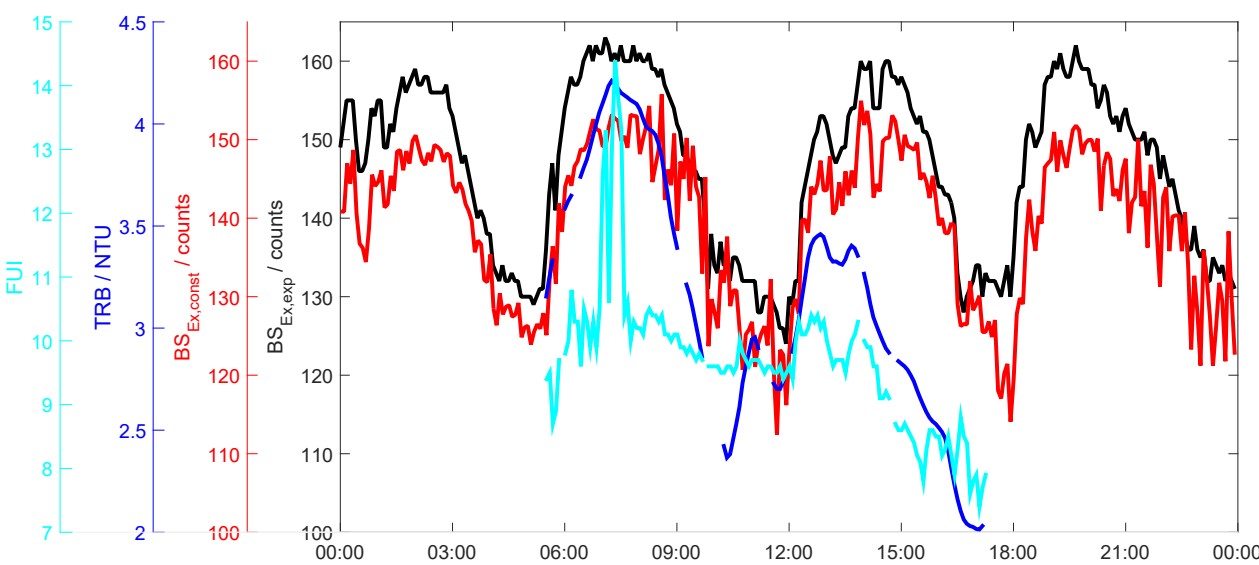

**Figure 7.** Time series observations on 29.08.2013 of Forel-Ule color index (FUI, cyan), backscatter signal (BS, constant extrapolation: red, exponential extrapolation: black) and turbidity (blue).





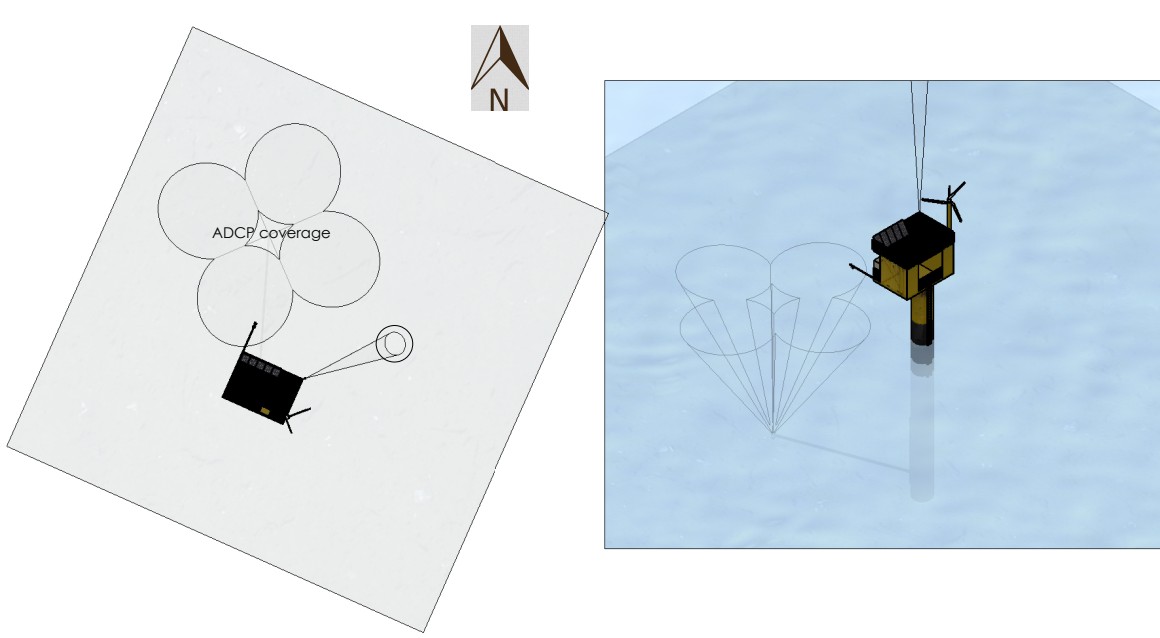

**Figure 8.** Schematic of the different measurement fields of view (FOV) of the sensors at the Time Series Station Spiekeroog at high water.
Left panel: top view; right panel: perspective from west (provided by Nick Rüssmeier).





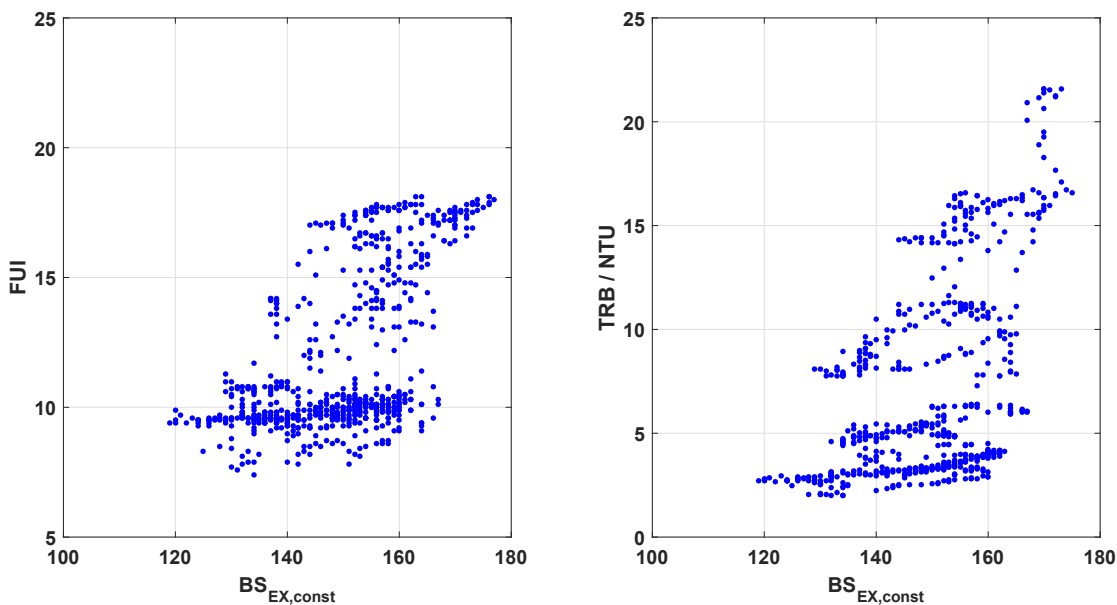

**Figure 9.** Comparison for the entire period (five days). Left: Forel-Ule-Index (FUI) and the backscatter signal $BS_{EX,const}$ . Right: turbidity (TRB) and the backscatter signal $BS_{EX,const}$ .





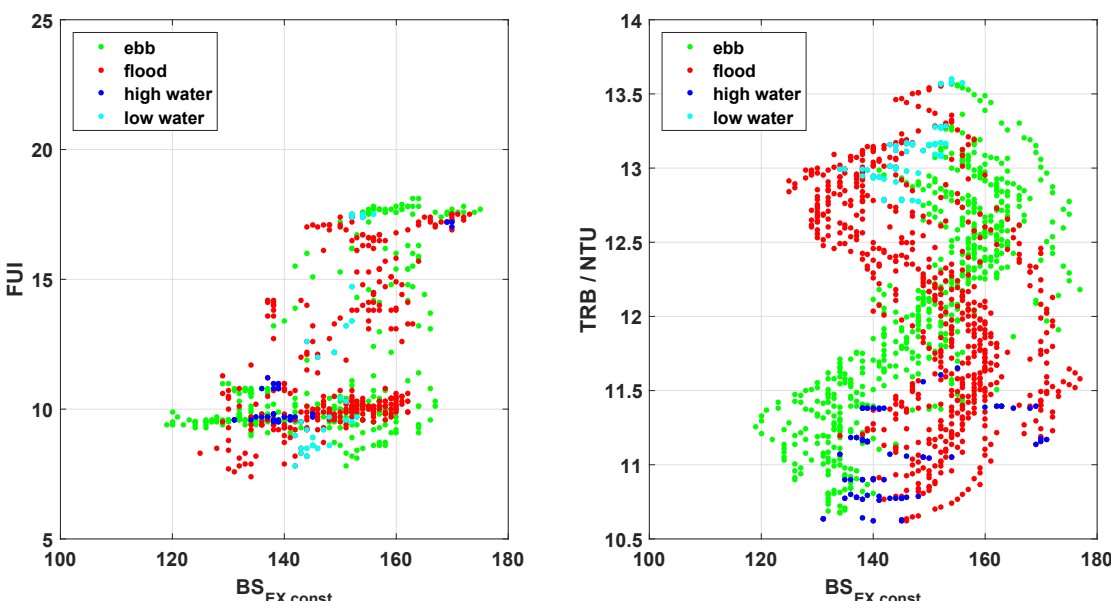

**Figure 10.** Comparison between backscatter signal $BS_{EX,const}$ and Forel-Ule-Index ($FUI$, left) and backscatter signal $BS_{EX,const}$ and turbidity data ($TRB$, right) separated by tidal phases; all averaged ebb values (green), all averaged flood values (red), all averaged high waters (dark blue), all averaged low waters (light blue). Data shown from 28 August 2013 until 02 September 2013.