# Peer review of "Acoustic and optical methods to infer water transparency at the Time Series Station Spiekeroog, Wadden Sea"

_Ocean Science, 2016_

## Referee Comment (RC1) · Anonymous Referee #1 · 30 Jun 2016

General comments The paper presents an interesting approach by comparing both optical and acoustical methods for obtaining information about the suspended sediments in the water, which is a key parameter for deriving water transparency. The study design is reasonable and the instrumental setup is well described. The introduction gives a good overview about the relevant topics, although some more information about the link between water transparency and the investigated optical proxies could be added (see also specific comments). However, in my opinion, especially the Results and Discussion section needs some revision before final publication. The data shown and arguments given are plausible and interesting, but the argumentation needs more focus. By now, it is relatively difficult to get some of the findings right on the first view,

especially the biofouling issue, which highlights the advantage of the reflectance and acoustic data. Hopefully, the following comments are helpful in this respect.

Specific comments

page 1, line 23: The optical properties of the water are also influenced by the dissolved and suspended matter present, so which additional factors play a role for determining water clarity/transparency? If it is only the dissolved and suspended matter, the reference to the optical properties is somewhat redundant and could be omitted.

page 2, line 1-11: Maybe it would be valuable to explain shortly how water color observations are linked to water transparency estimates. This would show the value of reflectance and Forel-Ule data more clearly.

page 3, line 28: How was the turbidity data quality-checked?

page 4, line 15: The polynomial fitting method is basically a linear fit, isn't it?

page 4, line 15: "The top right graphic shows the extrapolation results." This is confusing, because also in the top left figure there are the three results of the fits shown. The difference between left and right figure is the exclusion of data because of the bottom layer. Please rephrase the sentence.

page 4, line 23-24: Unfortunately, figure 5 confuses me a little bit. It is clear that you get an $R^2$ value for the correlation between the measured data and the fitted curves as a quality indicator of the respective fit. But I don't understand the histogram, where also $R^2$-values are shown. Have you repeated the respective fit several times and got different results? If this is the case, can you explain the reason? Otherwise, if the histogram has no special meaning and only the overall $R^2$-value is the important point, you should omit the histogram and give the $R^2$ values directly in the upper part of the figure (e.g. in the legend).

page 4, line 24: This is the only point in the manuscript where figure 6 is mentioned. The data shown are not used in the Results and Discussion section to explain certain

issues, so if it is not necessary, you should remove it.

page 4, line 27 to end of chapter: These paragraph should be moved to the Introduction, since it describes previous work, relationships between parameters and aims of the study.

page 5, line 3: Figure 8 shows the field of view of the sensors of the station. Maybe the figure should be mentioned and described the first time in the Materials and Methods section, since it is related to the measurement setup. Then it can be referred to in the Results and Discussion section.

Results and Discussion in general: At the beginning, the authors speak of a moderate correlation between the data shown in figure 7 observed by visual inspection. I assume that these are the same data which were used in table 1 for the one-day-correlation? If this is the case, please state this and refer also to these results at this point in the text to support your argumentation. Then there is no need to "confirm" correlations (line 19) afterwards, as they were already given and can be explained in the following. Generally, I would give the results of the Spearman-tests at the beginning of the chapter and then start to explain them (sediment types and different response of instruments). By now, it is the other way around. Furthermore, since the Forel-Ule data were already given in figure 7 and table 1, I would mention this method earlier in the section.

page 5 line 17: Do you mean the data shown in figure 7? If this is the case, why don't you give the tidal signal in the figure to support your statement?

page 5, line 26 - page 6, line 6: The differences between the correlations for the various tidal phases are shown, but not completely discussed. What are the reasons for the observed differences? The different kinds of sediments (fine, coarse) transported in the different tidal phases?

page 5, line 29: Is figure 9 the visual representation of two of the correlations shown in table 1? Why are explicitly these data shown again? Maybe this figure is redundant to

table 1?

page 5, line 31 - page 6, line 2: It is mentioned two times that figure 10 shows a comparison between backscatter and turbidity data. Please rephrase the sentence. Are the data shown also identical to the data used in table 2?

page 6, line 8-9: Where in the data or the Results and Discussion section has been shown the influence of biofouling? I assume it is the difference in the correlations between the one day period and the longer period (table 1). However, this should be clearly mentioned in the Results and Discussion, otherwise it is confusing why this statement is given in the conclusions and also in the abstract.

page 6, line 26-28: This sentence would be good at the beginning of the Conclusion section.

Technical corrections

page 2, line 4: "measurement" instead of measurements"

page 2, line 25: The last sentence of the paragraph could be better placed behind the sentence ending in line 22.

page 2, line 34: Please rephrase the sentence to "At the same time, this approach could contribute to an understanding of..."

page 3, line 5-6: The coordinates could be given without brackets.

page 3, line 9: The abbreviation TSS for Time Series Station Spiekeroog has already been introduced and could be used also here.

page 3, line 16: Maybe rather "Three radiometers continuously measure hyperspectral radiance and irridiance in 5 minute intervals to..."

page 3, line 21: Please rephrase to "...submerged ECO FLNTU sensor (WETlabs, USA)..."

page 4, line 14: "applied" instead of "implemented"

page 4, line 22: "whole" instead of "complete", because this is more related to the following abbreviation (wwc)

page 5, line 18: Please rephrase to "Thus, we presume these instruments provide reasonable proxies for the suspended material which are comparable."

page 5, line 20: "compare" instead of "compared"

Figure 1: The label "Turbidity Meter" should be placed closer the position of the instrument indicated in red. Furthermore, also the position of the radiometers should be indicated by red symbols to be consistent with the other instruments. Also the same font size should be used for all instrument labels.

Figure 2: The font size of the legend appears to be different. Furthermore, could you provide the units in e.g. brackets? Using a backslash gives the impression of a ratio on the first sight. This applies also to the other figures.

Figure 4: Line colors should be chosen according to the tidal phase to make the figure more clearly (e.g. red for flood, green for ebb etc.). To differentiate between two floods, for example, different shades of the respective color could be used or different types of lines.

Figure 8: Could you explain why the left square is turned? Also, the arrow indicating north should be given on the panel it belongs to.
* * *

---

## Referee Comment (RC2) · Anonymous Referee #2 · 3 Jul 2016

GENERAL COMMENTS: The paper presents a novel approach to comparison of different techniques for water transparency measurements using time series data. The purpose of this study was to find correlations between acoustic and optical measurements in order to fill possible gaps in water transparency data when individual instruments fail. This was successfully reached and important conclusions were made. Although, some parts of the text need clarification, I recommend this paper for publication in Ocean Science journal (Special Issue: COSYNA: integrating observations and modeling to understand coastal systems) after some minor revisions specified below.

SPECIFIC COMMENTS:

P2L25 explain 'SPM' at first mention

[Figure]

P3L18 explain NN

P3L19 It will be more clear if you give full reference of 2012, like in P2L10-11.

P3L28 How were these data checked for quality?

P3L29 How were they measured? Using the same instrument? Please precise.

P4L10-12 This sentence doesn't fit here, it should be moved elsewhere (it separates the information about extrapolation), e.g., in L4 before the information on Rmax. Also, there's no need to name plot colors in the text, they are visible on the graph.

P4L18 Try to make this description more clear. The sentence "The top right graphic shows the extrapolation results" is also true for the left graphic; try to highlight the difference between right and left graphics here, and afterwards explain. Is it the backscatter signal that is highly influenced by strong currents during ebb and flood? Be more precise. I guess, the currents may affect the ADCP backscatter signal as well as the presence of SPM. How do you separate this information? Describe or provide a reference.

P4L20-21 Can you provide a reference to support this assumption?

P4L22 Complete or whole? In the description of Fig. 5 you use 'whole' which seems better for the 'wwc' shortcut.

P4L24 I suggest to be more precise: 'acoustic backscatter', which 'other parameters' (they are only two, you can list them)?

P4L26 Support this assumption with a reference.

P4L27-28 This is a general statement appropriate for introduction. If you place it in 'Sampling and analysis' section, you should show how you applied this information in your study. Remove or reformulate.

P5L9 I would use 'scattering patterns' or 'scattering characteristics' instead of 'scatter

behaviour'; the difference is actually in applied method, not in the behaviour of sediments.

P5L15 I guess, d stands for 'diameter'. It should be mentioned in the text. In this sentence you speak about sediment concentration and dynamics, but the value you give in brackets is neither one nor the other. Please correct.

P5L18 'these' refers to what? It's not clear which instruments you speak about.

P5L22-24 You didn't explain here the difference between one day and longer period correlation. Why do you think the results are significantly better for one day data? Is it only for this specific day, or a general trend? More detailed explanation is advised here.

P5L26-27 This sentence is general, it could be good for introduction, but not in a discussion section.

P5L24-25 I have the impression that you chose the constant extrapolation only because it gave better results in your study. Can you try to provide a stronger scientific motivation based on other published research?

P5L29 I suggest to say more precisely 'ADCP backscatter signals' or 'acoustic backscatter signals'. It should be clear you don't refer to optical backscattering here.

P5L31-33 - P6L1-2 This figure description is not consistent. The first sentence speaks only about the right graph. In the second sentence 'also' seems not appropriate. Please check and reformulate this description.

P6L2-3 This information is repeated from P5L30. It is difficult to understand the idea of such repetition.

P6L8 This conclusion seems to appear out of nowhere. It is not the result of your study. It seems to be your conclusion/assumption, but in the results and discussion section you don't speak about the possible artefacts of bio-fouling. I suggest to add a

short description about the possible influence of bio-fouling on the quality of turbidity data. How long after or before cleaning maintenance was your 5-day study period? Do you assume that the one day study period was less influenced by bio-fouling? Did you receive similar conclusions in your earlier studies quoted in this paper, where you linked turbidity to Forel-Ule index?

P6L11 'is' seems to strong in this place; I would say 'can be' - looking at the correlation test results.

P6L12-13 This sentence should be in discussion section, and the results of Schultz et al. (2015) should be described briefly in comparison to your current results.

P6L13 Be careful with conclusions that go too far. Which responses do you call linear in your study? Spearman rank test doesn't prove linear correlation, but a monotonic correlation (which can be non-linear). Only some of your results show very strong or strong correlation; most of correlations is moderate or weak. I suggest simply to show your best and worse result with appropriate comments.

P6L18 Avoid citation in conclusions.

P6L20 powerful tool to do what? Be more precise.

Citations: 17 out of 41 references are authors' self-citations (which is more than 40%). I have the impression that some paper are quoted unnecessarily, giving three or four references to support one thesis is too much. If you see their findings are essential for your study, point them precisely in separate sentences.

All figures: the units in text are presented in a rectangular brackets. Please use the same way on figures and their descriptions.

Fig. 3. Description: add short information about the device and location

Fig. 4. The same. Can you add another LW profile plot? The graph makes a reader wonder why all examples are double except of LW. Height and depth - choose one way

to describe this quantity.

Fig. 3-7,10-11. I suggest to add 'acoustic' or 'ADCP' backscatter in the figures' description.

Fig. 8. description: 'schematic' or 'scheme'?

Why the right graph of Fig. 9 contains much less points than the corresponding graph of Fig. 10? Both figures' descriptions show the same five-day period. Please explain and/or correct.

LANGUAGE COMMENTS: Although English is not my first language I see some minor grammar, punctuation and syntax errors. I recommend to use a professional English correction service, and in particular please check the following sentences/phrases:

P1L2-3 I suggest to continue the sentence in past tense: determined, demonstrated

P1L4 is=comes

P4L4 ...one of such...

P2L20 comma needed

P3L29 can be?

P4L8 "between ... and ..." or "from ... to ..."

P4L34 'earlier' or 'previous'

P5L20 compare

P6L13 sensor

P6L14 agree

P6L26 was=were

---

## Author Comment (AC1) · 7 Sep 2016

First of all we want to thank both reviewers for their time that they have taken to improve and comment this publication.

Blue – Author Response

Green – Manuscript text revised by authors

**Anonymous Referee #1**

**General comments**

The paper presents an interesting approach by comparing both optical and acoustical methods for obtaining information about the suspended sediments in the water, which is a key parameter for deriving water transparency. The study design is reasonable and the instrumental setup is well described. The introduction gives a good overview about the relevant topics, although some more information about the link between water transparency and the investigated optical proxies could be added (see also specific comments). However, in my opinion, especially the Results and Discussion section needs some revision before final publication. The data shown and arguments given are plausible and interesting, but the argumentation needs more focus. By now, it is relatively difficult to get some of the findings right on the first view, especially the biofouling issue, which highlights the advantage of the reflectance and acoustic data. Hopefully, the following comments are helpful in this respect. Thank you for the positive feedback. Based on the two reviewers comments the manuscript was thoroughly revised and improved. We split up and rearranged the results and discussion section in two separate sections for a better structure and overview. Further, we focused more on the description of the results and especially pointing out the bio-fouling issue.

**Specific comments**

page 1, line 23: The optical properties of the water are also influenced by the dissolved and suspended matter present, so which additional factors play a role for determining water clarity/transparency? If it is only the dissolved and suspended matter, the reference to the optical properties is somewhat redundant and could be omitted.

The sentence intended to describe that the color and transparency is influenced by dissolved material, suspended material and the water itself (its scattering and absorption properties). We omitted the sentence since we rearranged the introduction.

page 2, line 1-11: Maybe it would be valuable to explain shortly how water color observations are linked to water transparency estimates. This would show the value of reflectance and Forel-Ule data more clearly.

A decrease of water clarity (resulting in lower Secchi depths) and therefore water transparency typically goes along with a shift to higher Forel-Ule-Colour indices. We rearranged the introduction to provide focus on the relevant parameters of this study.

page 3, line 28: How was the turbidity data quality-checked?

The quality control protocol is presented in the JEOS paper Garaba et al. (2014), it was an arbitrary threshold set based on mean turbidity changes just after cleaning. We explicitly added the link to the Garaba et al. (2014) paper to avoid doubling yet providing all necessary information to the reader.

page 4, line 15: The polynomial fitting method is basically a linear fit, isn't it?

Yes, it is a linear (first order polynomial) fit. We added that information to the manuscript.

page 4, line 15: "The top right graphic shows the extrapolation results." This is confusing, because also in the top left figure there are the three results of the fits shown. The difference between left and right figure is the exclusion of data because of the bottom layer. Please rephrase the sentence.

We changed the manuscript accordingly.

The top graphics show an exemplary 5 profile during low water and one during flood (violet and blue in Fig. 3) with different extrapolations to the surface layer. The top graphics distinguish with the range at which the extrapolation is applied: The extrapolation through the whole water column is shown in the left column. The right graphic shows the extrapolation applied on data where no influence through bottom friction is. Here, the data derived in the near bottom range up to 4 m are excluded since it is highly influenced by strong currents during ebb and flood.

page 4, line 23-24: Unfortunately, figure 5 confuses me a little bit. It is clear that you get an $R^2$ value for the correlation between the measured data and the fitted curves as a quality indicator of the respective fit. But I don't understand the histogram, where also $R^2$-values are shown. Have you repeated the respective fit several times and got different results? If this is the case, can you explain the reason? Otherwise, if the histogram has no special meaning and only the overall $R^2$-value is the important point, you should omit the histogram and give the $R^2$ values directly in the upper part of the figure (e.g. in the legend).

We applied the different extrapolation methods to the entire data set. For every acoustic backscatter profile, we got a corresponding $R^2$ value. To evaluate which of the methods would be best, we made histograms of all the resulting $R^2$ values.

page 4, line 24: This is the only point in the manuscript where figure 6 is mentioned. The data shown are not used in the Results and Discussion section to explain certain issues, so if it is not necessary, you should remove it.

We removed the figure and changed the manuscript accordingly.

page 4, line 27 to end of chapter: These paragraph should be moved to the Introduction, since it describes previous work, relationships between parameters and aims of the study.

We changed the manuscript accordingly.

page 5, line 3: Figure 8 shows the field of view of the sensors of the station. Maybe the figure should be mentioned and described the first time in the Materials and Methods section, since

it is related to the measurement setup. Then it can be referred to in the Results and Discussion section.

We changed the manuscript accordingly.

**Results and Discussion in general:**

At the beginning, the authors speak of a moderate correlation between the data shown in figure 7 observed by visual inspection. I assume that these are the same data which were used in table 1 for the one-day-correlation?

Yes, that is right

If this is the case, please state this and refer also to these results at this point in the text to support your argumentation. Then there is no need to "confirm" correlations (line 19) afterwards, as they were already given and can be explained in the following.

We changed the manuscript accordingly.

Generally, I would give the results of the Spearman-tests at the beginning of the chapter and then start to explain them (sediment types and different response of instruments). By now, it is the other way around. Furthermore, since the Forel-Ule data were already given in figure 7 and table 1, I would mention this method earlier in the section.

We rearranged the results section and separated it from the discussion section improving the logical flow of manuscript however we stick to the line of argument that we found more appropriate in this case.

page 5 line 17: Do you mean the data shown in figure 7? If this is the case, why don't you give the tidal signal in the figure to support your statement?

Yes. We added the water level as a variable which represent the tidal signal and describe it in the text.

page 5, line 26 - page 6, line 6: The differences between the correlations for the various tidal phases are shown, but not completely discussed. What are the reasons for the observed differences? The different kinds of sediments (fine, coarse) transported in the different tidal phases?

One difference causes probably from the different kinds of sediments. At the moment we have only modelling data about the sediment distribution and dynamics. For a detailed and qualified statement we have to measure the particles size from sampling at that location over a tidal cycle and longer. Thank you for this comment, we will keep this in mind for a future field campaign.

page 5, line 29: Is figure 9 the visual representation of two of the correlations shown in table 1? Why are explicitly these data shown again? Maybe this figure is redundant to table 1?

Yes, it is redundant. We removed that figure.

page 5, line 31 - page 6, line 2: It is mentioned two times that figure 10 shows a comparison between backscatter and turbidity data. Please rephrase the sentence. Are the data shown also identical to the data used in table 2?

We changed the manuscript accordingly. Yes, the figure is redundant. We removed it.

page 6, line 8-9: Where in the data or the Results and Discussion section has been shown the influence of biofouling? I assume it is the difference in the correlations between the one day

period and the longer period (table 1). However, this should be clearly mentioned in the Results and Discussion, otherwise it is confusing why this statement is given in the conclusions and also in the abstract.

It is shown in Fig. 2. We explained it in more detail in the text.

Turbidity time series (Fig. 2) shows a rapid response directly after cleaning of the ECO FLNTU sensor as expected in a highly bio-active season (summer). Even in this short time period of 6 days a strong increase and spreading of turbidity values is discernible. Directly after cleaning maximum values are below 5 NTU with a range of 2.2 NTU, already after 3 days the increase started, at the end of this 6-day-period the values spread out to 10 NTU and reach the upper range of the reasonable data for turbidity at nearly 25 NTU.

page 6, line 26-28: This sentence would be good at the beginning of the Conclusion section.

We changed the manuscript accordingly.

**Technical corrections**

We changed the manuscript for all following comments accordingly.

page 2, line 4: "measurement" instead of measurements"

page 2, line 25: The last sentence of the paragraph could be better placed behind the sentence ending in line 22.

page 2, line 34: Please rephrase the sentence to "At the same time, this approach could contribute to an understanding of..."

page 3, line 5-6: The coordinates could be given without brackets.

page 3, line 9: The abbreviation TSS for Time Series Station Spiekeroog has already been introduced and could be used also here.

page 3, line 16: Maybe rather "Three radiometers continuously measure hyperspectral radiance and irridiance in 5 minute intervals to..."

page 3, line 21: Please rephrase to "...submerged ECO FLNTU sensor (WETlabs,

USA)..."

page 4, line 14: "applied" instead of "implemented" page 4, line 22: "whole" instead of "complete", because this is more related to the following abbreviation (wwc) page 5, line 18: Please rephrase to "Thus, we presume these instruments provide reasonable proxies for the suspended material which are comparable." page 5, line 20: "compare" instead of "compared"

Figure 1: The label "Turbidity Meter" should be placed closer the position of the instrument indicated in red. Furthermore, also the position of the radiometers should be indicated by red symbols to be consistent with the other instruments. Also the same font size should be used for all instrument labels.

Figure 2: The font size of the legend appears to be different. Furthermore, could you provide the units in e.g. brackets? Using a backslash gives the impression of a ratio on the first sight. This applies also to the other figures.

Figure 4: Line colors should be chosen according to the tidal phase to make the figure more clearly (e.g. red for flood, green for ebb etc.). To differentiate between two floods, for example, different shades of the respective color could be used or different types of lines.

Figure 8: Could you explain why the left square is turned? Also, the arrow indicating north should be given on the panel it belongs to.

The left panel is turned because it belong to the arrow which indicates the north.

**Anonymous Referee #2**

**GENERAL COMMENTS:**

The paper presents a novel approach to comparison of different techniques for water transparency measurements using time series data. The purpose of this study was to find correlations between acoustic and optical measurements in order to fill possible gaps in water transparency data when individual instruments fail. This was successfully reached and important conclusions were made. Although, some parts of the text need clarification, I recommend this paper for publication in Ocean Science journal (Special Issue: COSYNA: integrating observations and modeling to understand coastal systems) after some minor revisions specified below.

Thank you for the very encouraging and positive feedback.

**SPECIFIC COMMENTS:**

P2L25 explain 'SPM' at first mention

We changed the manuscript accordingly.

P3L18 explain NN

NN is the german abbreviation on the mean sea level, which is a national reference height in the geographical height system. Was changed.

P3L19 It will be more clear if you give full reference of 2012, like in P2L10-11.

Yes, you are right. We rewrite and reduce the paragraph.

Hyperspectral radiometers were used to collect and derive remote sensing reflectance ($R_{RS}$) measurements at 24 m above the seafloor at 5 minute interval continuously. The reflectance measurements, corrected for environmental perturbations, were transformed into Forel-Ule color indices that can be matched to the intrinsic color of water. A submerged WETlabs ECO

FLNTU sensor measures turbidity at 12 m above the seafloor at 1 minute interval continuously. The ECO FLNTU sensor samples turbidity data with optical backscattering at a wavelength of 700 nm. Detailed information on the processing of these measurements is presented in an earlier study Garaba et al. (2014).

P3L28 How were these data checked for quality?

The quality control protocol is presented in the JEOS paper Garaba et al. (2014), it was an arbitrary threshold set based on mean turbidity changes just after cleaning. We added an explicit link to this reference in the manuscript.

P3L29 How were they measured? Using the same instrument? Please precise.

These turbidity data were measured with the ECO FLNTU sensor near the surface of the water column at a fixed height above the seabed (see Garaba et al. 2014).

P4L10-12 This sentence doesn't fit here, it should be moved elsewhere (it separates the information about extrapolation), e.g., in L4 before the information on Rmax. Also, there's no need to name plot colors in the text, they are visible on the graph.

We changed the manuscript accordingly.

In different tidal phases (flood, ebb, slack water) the acoustic backscatter signal shows different profiles over depth (Fig. 3). Depending on the current velocity (e.g. at slack water or at maximum velocity), the profiles have similar shapes.

P4L18 Try to make this description more clear. The sentence "The top right graphic shows the extrapolation results" is also true for the left graphic; try to highlight the difference between right and left graphics here, and afterwards explain. Is it the backscatter signal that is highly influenced by strong currents during ebb and flood? Be more precise. I guess, the currents may affect the ADCP backscatter signal as well as the presence of SPM. How do you separate this information? Describe or provide a reference.

The amount of particles within the water column depends on the currents, therefore the backscatter signal is indirectly influenced by the currents.

We changed the manuscript accordingly.

The top graphics show an exemplary profile during low water and one during flood (violet and blue in Fig. 3) with different extrapolations to the surface layer. The top graphics distinguish with the range at which the extrapolation is applied: The extrapolation through the whole water column is shown in the left column. The right graphic shows the extrapolation applied on data where no influence through bottom friction is. Here, the data derived in the near bottom range up to 4 m are excluded since it is highly influenced by strong currents during ebb and flood.

P4L20-21 Can you provide a reference to support this assumption?

Investigations from e.g. Badewien et al. (2009) and van der Hout et al. (2015) showed partially strong vertical gradients in SPM in coastal areas with a reduced variation in the surface concentration signal.

P4L22 Complete or whole? In the description of Fig. 5 you use 'whole' which seems better for the 'wwc' shortcut.

We changed the manuscript accordingly.

P4L24 I suggest to be more precise: 'acoustic backscatter', which 'other parameters' (they are only two, you can list them)?

We changed the manuscript accordingly.

P4L26 Support this assumption with a reference.

See answer for P4L22.

P4L27-28 This is a general statement appropriate for introduction. If you place it in 'Sampling and analysis' section, you should show how you applied this information in your study. Remove or reformulate.

We removed the sentence to the introduction.

P5L9 I would use 'scattering patterns' or 'scattering characteristics' instead of 'scatter behaviour'; the difference is actually in applied method, not in the behaviour of sediments.

We changed the manuscript accordingly.

P5L15 I guess, d stands for 'diameter'. It should be mentioned in the text. In this sentence you speak about sediment concentration and dynamics, but the value you give in brackets is neither one nor the other. Please correct.

We changed the manuscript accordingly.

P5L18 'these' refers to what? It's not clear which instruments you speak about.

We changed the manuscript accordingly.

P5L22-24 You didn't explain here the difference between one day and longer period correlation. Why do you think the results are significantly better for one day data? Is it only for this specific day, or a general trend? More detailed explanation is advised here.

The bio-fouling influence started already in the short six-day time period, therefore correlation values of the shorter one-day time period directly after cleaning of the ECO FLNTU sensor were stronger than the values for the entire time period of six days.

P5L26-27 This sentence is general, it could be good for introduction, but not in a discussion section.

We changed the manuscript accordingly.

P5L24-25 I have the impression that you chose the constant extrapolation only because it gave better results in your study. Can you try to provide a stronger scientific motivation based on other published research?

For further investigations, we used the constant extrapolated acoustic backscatter signal $BS_{Ex,const.}$ This approach corresponded to our assumption of homogeneity of the surface layer.

P5L29 I suggest to say more precisely 'ADCP backscatter signals' or 'acoustic backscatter signals'. It should be clear you don't refer to optical backscattering here.

We changed the manuscript accordingly.

P5L31-33 - P6L1-2 This figure description is not consistent. The first sentence speaks only about the right graph. In the second sentence 'also' seems not appropriate. Please check and reformulate this description.

We changed the manuscript accordingly.

P6L2-3 This information is repeated from P5L30. It is difficult to understand the idea of such repetition.

We changed the manuscript accordingly.

P6L8 This conclusion seems to appear out of nowhere. It is not the result of your study. It seems to be your conclusion/assumption, but in the results and discussion section you don't speak about the possible artefacts of bio-fouling. I suggest to add a short description about the possible influence of bio-fouling on the quality of turbidity data. How long after or before cleaning maintenance was your 5-day study period? Do you assume that the one day study period was less influenced by bio-fouling? Did you receive similar conclusions in your earlier studies quoted in this paper, where you linked turbidity to Forel-Ule index?

We changed the manuscript accordingly.

Turbidity time series (Fig. 2) shows a rapid response directly after cleaning of the ECO FLNTU sensor as expected in a highly bio-active season (summer). Even in this short time period of 6

days a strong increase and spreading of turbidity values is discernible. Directly after cleaning maximum values are below 5 NTU with a range of 2.2 NTU, already after 3 days the increase started, at the end of this 6-day-period the values spread out to 10 NTU and reach the upper range of the reasonable data for turbidity at nearly 25 NTU.

P6L11 'is' seems to strong in this place; I would say 'can be' - looking at the correlation test results.

We changed the manuscript accordingly.

P6L12-13 This sentence should be in discussion section, and the results of Schultz et al. (2015) should be described briefly in comparison to your current results.

We changed the manuscript accordingly.

Our results regarding the correlation between acoustic backscatter signal and turbidity agree well with investigations of Schulz et al. (2015). The data sets of the in-water sensors showed moderate to strong correlations, especially the counter wise strengths of the signals during the tidal signal are found again.

P6L13 Be careful with conclusions that go too far. Which responses do you call linear in your study? Spearman rank test doesn't prove linear correlation, but a monotonic correlation (which can be non-linear). Only some of your results show very strong or strong correlation; most of correlations is moderate or weak. I suggest simply to show your best and worse result with appropriate comments.

We changed the manuscript accordingly.

P6L18 Avoid citation in conclusions.

We changed the manuscript accordingly.

P6L20 powerful tool to do what? Be more precise.

We changed the manuscript accordingly.

On a qualitative level, using the Forel-Ule-Index, as derived from radiometer measurements, is a powerful tool for exchangeable estimations of water transparency as much as data sets derived from ADCP measurements.

Citations: 17 out of 41 references are authors' self-citations (which is more than 40%). I have the impression that some paper are quoted unnecessarily, giving three or four references to support one thesis is too much. If you see their findings are essential for your study, point them precisely in separate sentences.

This paper is based on a series of dedicated research activities from the authors over the last decade and it is part of a special issue on COSYNA summing up the work of a multi-year project. It is therefore our aim to link the advances presented to this body of work and provide the reader with the appropriate references.

To prevent overloading and doubling, we checked the references and indicated them at the specific sentences. In total the number of references and self-citations was reduced.

**All figures**: the units in text are presented in a rectangular brackets. Please use the same way on figures and their descriptions.

We changed the figures accordingly.

Fig. 3. Description: add short information about the device and location Fig. 4. The same. Can you add another LW profile plot? The graph makes a reader wonder why all examples are double except of LW. Height and depth - choose one way to describe this quantity.

We changed the description and figures accordingly.

Fig. 3-7,10-11. I suggest to add 'acoustic' or 'ADCP' backscatter in the figures' description.

We changed the descriptions accordingly.

Fig. 8. description: 'schematic' or 'scheme'?

We decided to choose 'schematic'.

Why the right graph of Fig. 9 contains much less points than the corresponding graph of Fig. 10? Both figures' descriptions show the same five-day period. Please explain and/or correct.

We selected time periods which we can assign to the phases of the tidal signal and averaged them.

**LANGUAGE COMMENTS:**

We changed the manuscript for all following comments accordingly and had an expert English speaker additionally reviewing the manuscript.

Although English is not my first language I see some minor grammar, punctuation and syntax errors. I recommend to use a professional English correction service, and in particular please check the following sentences/phrases:

P1L2-3 I suggest to continue the sentence in past tense: determined, demonstrated

P1L4 is=comes

P4L4 ...one of such... P2L20 comma needed P3L29 can be?

P4L8 "between ... and ..." or "from ... to ..."

P4L34 'earlier' or 'previous'

P5L20 compare

P6L13 sensor

P6L14 agree

P6L26 was=were